# Fungal Priming: Prepare or Perish

**DOI:** 10.3390/jof8050448

**Published:** 2022-04-25

**Authors:** Ety Harish, Nir Osherov

**Affiliations:** Department of Clinical Microbiology and Immunology, Sackler School of Medicine, Tel-Aviv University, Tel-Aviv 69978, Israel; etyharish@mail.tau.ac.il

**Keywords:** fungal priming, acclimation, acquired stress resistance, adaptive response, cross-protection

## Abstract

Priming (also referred to as acclimation, acquired stress resistance, adaptive response, or cross-protection) is defined as an exposure of an organism to mild stress that leads to the development of a subsequent stronger and more protective response. This memory of a previously encountered stress likely provides a strong survival advantage in a rapidly shifting environment. Priming has been identified in animals, plants, fungi, and bacteria. Examples include innate immune priming and transgenerational epigenetic inheritance in animals and biotic and abiotic stress priming in plants, fungi, and bacteria. Priming mechanisms are diverse and include alterations in the levels of specific mRNAs, proteins, metabolites, and epigenetic changes such as DNA methylation and histone acetylation of target genes.

## 1. Introduction

This review provides an introduction to priming in the kingdoms of life, with an emphasis on fungal priming. First, we describe the extensive priming studies performed in the yeast *Saccharomyces cerevisiae*, and compare them to the findings in *Candida albicans,* a leading human fungal pathogen. Next, we characterize priming in the filamentous fungi, including the aseptate mucorales *Rhizopus arrhizus*, the insect pathogen *Metarhizium anisopliae*, the plant pathogen *Penicillium chrysogenum*, and the human pathogen *Aspergillus fumigatus*. Recent exciting findings demonstrate that priming occurs not only in the growing fungal mycelium but also within spores. This allows the organism to release large numbers of propagules that are pre-programmed to better survive under the adverse conditions in which they were formed. Priming conditions that increase spore virulence or resistance to antifungal drugs are of important medical and agricultural relevance. Understanding the mechanisms responsible for priming in filamentous fungi remains a major unexplored frontier.

What is priming? There is growing evidence that animals, plants, bacteria, and fungi can ‘remember’ a past experience. The memory of a past event may shape or ‘prime’ their response to future external stressors, resulting in a subsequent robust response. Priming (also referred to as acclimation, acquired stress resistance, adaptive response, or cross-protection) is defined as a time-limited pre-exposure of an organism to stress that leads to an increased adaptive response to subsequent exposures [1]. The initial priming stress can be identical or different from the subsequent exposure. If the initial priming stress and the subsequent exposure are of the same nature, they are referred to as *cis*-priming. If priming and exposure differ, for example, heat priming followed by exposure to oxidative stress, they are termed *trans*-priming [1]. The duration of the memory resulting from the priming stress event can vary from days to weeks, and even in some cases, can be inherited by the offspring. Intergenerational priming occurs when the stress memory is observed in the first, stress-free offspring generation, whereas trans-generational priming is observed after more than two stress-free offspring generations [1,2,3]. Priming mechanisms are not well understood but appear to involve epigenetic, cellular, and other non-genetic mechanisms [3] (Figure 1).

Priming has been studied in vertebrates, invertebrates, plants, and bacteria [1,2,4,5,6,7,8] (Table 1). Different types of priming have been described in vertebrates and invertebrates, including immune priming [5,9] and transgenerational priming [5,6,10,11,12,13,14,15,16,17]. Priming in plants occurs primarily as a result of abiotic (e.g., drought, low and high temperatures, osmotic stress) [18,19,20,21,22] and biotic stress (e.g., fungi, viruses, bacteria, and hormones) [23,24,25], and is mainly driven by epigenetic mechanisms [2,3]. Priming in bacteria has been observed in response to both abiotic (temperature, hypoxia) [26,27] and biotic stress (antibiotics, antifungal peptides) [28,29,30,31,32], with cross resistance to biotic stress sometimes occurring as a result of abiotic stress [29,30].

This review focuses primarily on fungal priming, particularly in filamentous fungi. The field of priming in filamentous fungi is in its infancy, providing fertile ground for innovative research. Moreover, fungal priming may have a potentially paradigm-shifting impact in the fields of ecology, crop infections, and human disease.

## 2. Priming in Fungi

In fungi, priming has been most studied in the model yeast *Saccharomyces cerevisiae* and reflects a general homeostatic stress response [4,37]. In *S. cerevisiae*, the contribution of heat-shock priming to acquired stress resistance to subsequent severe stressors (trans-priming) was described in several studies [38,39,40] (Table 2). For example, *S. cerevisiae* primed by heat shock (37 °C, 1 h) or osmotic stress (0.7 M, 1 h) displayed tolerance to a following heat shock stress (48 °C). However, heat shock stressed cells were not tolerant against a following exposure to acute osmotic stress (1.5 M) [40]. These findings were reinforced by the study conducted by Coote et al., where priming of *S. cerevisiae* with several sub-lethal temperatures increased yeast thermotolerance to a higher temperature (52 °C) [38]. Yeast exposed to near freezing temperatures show increased tolerance for subsequent exposure to low temperatures and freezing. This response involves Msn2p and Msn4p transcriptional activation of the trehalose synthesizing genes *TPS1* and *TPS2*, leading to accumulation of protective trehalose [41]. Prior exposure of *S*. *cerevisiae* to ethanol (6%) and sorbic acid (9 M) resulted in increased thermotolerance to the same subsequent heat shock stress (52 °C), suggesting that acquired thermotolerance can result from a general stress response that is not specifically caused by the same priming agent [37,38]. Heat shock or ethanol priming protects *S. cerevisiae* against subsequent exposure to oxidative stress, emulating the adaptive responses found in the natural habitat of this yeast [42]. These findings suggest that different priming responses in *S. cerevisiae* share common genetic elements. Indeed, Hsp104p, responsible for disassembling protein aggregates, was strongly induced following priming by both heat and ethanol stress. Similarly, the disaccharide trehalose is strongly induced by heat, ethanol, and oxidative stress priming, functioning as a chemical protein-folding chaperone, free radical scavenger, and stabilizer of phospholipid membranes [37]. Mutational and transcriptome analysis revealed that the “general stress” transcription factors Msn2p and Msn4p [43], as well as the transcription factor Mga2 [44], involved in fatty acid biosynthesis, ergosterol biosynthesis, and the response to hypoxia, are required for heat, salt, and oxidative stress priming. Likewise, regulatory cross-talk between the transcription factors Msn2p, Msn4p, and Pdr3p was seen following priming by heat, organic acids, and osmotic stress [37]. Priming in yeast requires nascent protein synthesis during the pre-exposure step [43] and can persist for up to five generations, suggesting an epigenetic mechanism [39].

In contrast to the “general stress” response in *S. cerevisiae*, the pathogenic yeast *Candida albicans* does not show strong cross-protection to a subsequent stressor which is different from the pretreatment [56]. For example, pretreatment of *C. albicans* with mild heat shock only slightly increased its resistance to oxidative stress, and mild stress, induced by osmotic or oxidative stress, did not improve the survival rate of the yeast when a subsequent heat shock stress was applied [46] (Table 2).

Although the general stress response and adaptation to several specific stressors differ between *S. cerevisiae* and *C. albicans* [56], both fungi were highly sensitive to glucose levels, a fact which may affect their response to a following stress stimulus. Rodaki et al. demonstrated that in *C. albicans*, stress-related genes such as TPS1, TPS2, and TPS3 involved in the biosynthesis of trehalose were upregulated upon priming with low concentrations of glucose [47]. This upregulation helped the fungus better cope with subsequent oxidative and cationic stress and also contributed to azole antifungal resistance, indicating that cross-protection occurs in *C. albicans*. Interestingly, glucose treatment in *S. cerevisiae* leads to down-regulation of stress response genes [47]. The differences between *S. cerevisiae* and *C. albicans* cross-resistance were further investigated by Enjalbert et al., who revealed that *C. albicans* has a core stress response, but one that contains a smaller subset of genes [57], including the stress-activated protein Hog1 and the transcription factor Cap1 which have diverged significantly from *S. cerevisiae* [56,57,58]. In addition, homologs of the *S. cerevisiae* transcription factors Msn2 and Msn4, involved in regulating its core stress response, do not play equivalent roles in *C. albicans* [59]. These molecular differences between the two yeasts, resulting from their radically different lifestyles, may explain their different priming responses and cross-resistance to various stressors.

## 3. Priming in the Filamentous Fungi

In filamentous fungi, the phenomenon of priming has been understudied and remains generally descriptive [4] (Table 2). Priming has been described following exposure to several exogenous stresses, such as insect grazing, drugs, shear stress, or high temperatures [51,52,60]. Temperature priming was demonstrated in several species of Ascomycetes and Mucorales soil fungi, indicating that priming phenomena are likely widespread [61]. The duration of priming memory was tested in a recent study in two filamentous fungi. *Neurospora crassa* and *Penicillium chrysogenum* were primed by drought, and the priming response duration or memory was measured [50]. Both fungi were exposed to mild drought, then allowed to recover for 1, 7, or 14 d, then triggered again with severe drought. Following this stress, the *P. chrysogenum*-primed conidia showed improved growth and metabolic activity compared to unprimed conidia for up to 14 days, whereas in *N. crassa*, no priming responses were observed. This suggests that unlike in yeast, the priming-induced memory in some filamentous fungi can last for weeks.

In another study, *Metarhizium anisopliae* conidia were exposed to several different stressors (heat stress, oxidative stress, osmotic stress, and nutritive stress) at sub-lethal concentrations [48]. Results showed that both starvation-primed conidia and osmotic-primed conidia exhibited higher tolerance to a sequential stressor such as UV-B radiation or heat. In contrast, oxidative-stress-primed conidia did not show increased tolerance to UV-B or heat in the subsequent exposure. Heat-shock stressed conidia exhibited slightly but still increased tolerance to both the subsequent stressors of UV-B radiation or heat stress. The conidia were tested for sugar content to identify the priming mechanism. Increased protective trehalose and mannitol levels were found in conidia produced under nutritive stress, but not in conidia produced under osmotic stress. These findings indicate that the induction of trehalose and mannitol are not the only mechanism underlying the priming response in *M. anisopliae*.

The filamentous fungus *Rhizopus arrhizus*, which belongs to the Mucorales division of fungi and is known to cause life-threatening infection in immunocompromised patients, was also studied for priming. *R. arrhizus* spores were primed for 30 min by magnetic stirring to mimic tornadic shear stress. Then, the tornadic shear-primed spores were injected into *Drosophila melanogaster* fruit flies, and the survival rate of the flies was determined. The flies injected with shear-primed spores exhibited significantly higher mortality than those injected with unprimed spores. The priming effect observed in this fungus was attributed to both secreted metabolites from the fungus and activation of the calcineurin-signaling pathway [52]. Similarly, increased *R. arrhizus* virulence in *D. melanogaster* was also seen after priming with the medical triazole voriconazole [51].

A recent study conducted by Guhr et al. used the antioxidant organic compound riboflavin (vitamin B2) as a priming agent for the edible basidiomycete mushroom *Agaricus bisporus*, to examine the fungal response to drought after the application of riboflavin [49]. Results showed that riboflavin application attenuated the drought-induced stress phenotype. Although the exact mechanism was not characterized in this study, the application of riboflavin altered gene expression, suggesting epigenetic mechanisms underlying this priming effect. This is an example of a priming agent which is not an abiotic stress, but rather a substance that confers protection against fungal dehydration.

In summary, little is known about the priming responses of filamentous fungi to osmotic, nutritive, cold, and other abiotic and biotic stressors. Moreover, priming may affect fungi not only at the individual level but also at the community level, since in nature the growth of isolated filamentous fungi is rare [62]. Therefore, understanding priming may also contribute to better understanding the dynamics between filamentous fungi.

## 4. Priming in Aspergilli

Several recent studies have described interesting examples of priming in the Aspergilli (Table 2). Doll et al. studied the priming effect of fungivore grazing by *Folsomia candida* on *Aspergillus nidulans* [63]. Fungivore-exposed colonies of *A. nidulans* produced significantly higher amounts of toxic secondary metabolites and higher ascospore production relative to unchallenged fungi.

Hagiwara et al. investigated the effects of temperature during conidiation on the stress resistance of *A. fumigatus* conidia [53]. Conidia generated under elevated temperature priming showed a higher tolerance for heat, oxidative stress, and UV radiation. This was accompanied by increased trehalose levels, which may protect against these stresses. In another study, *A. fumigatus* was primed under nine different environmental conditions [54]. Each condition generated conidia with different germination and growth rates. For example, conidia generated under osmotic stress priming grew faster under various conditions. In contrast, conidia primed under metal deficiency exhibited slower germination and growth. Priming at 50 °C generated larger conidia and killed *G. mellonella* larvae faster. The priming mechanism underlying the hypervirulence of these conidia remains unknown.

In a landmark study, Wang et al. showed that conidia of the two filamentous fungi, *A. fumigatus* and *A. nidulans* remained transcriptionally active as long as they remained attached to the conidiophore. This is a surprising finding, as it was previously believed that mature conidia attached to the conidiophore were transcriptionally dormant [55]. Wang et al. showed that attached conidia exposed to various stresses underwent specific transcriptional changes and were more resistant to subsequent exposure to the same (cis-priming) or different (trans-priming) stresses. For example, priming with NaCl, generated conidia with higher levels of ergosterol pathway transcripts and increased tolerance to azole antifungals. Interestingly, it also increased the virulence of the conidia, following infection into *Galleria mellonella* moth larvae. Priming under conditions of zinc starvation generated conidia with high levels of gliotoxin biosynthesis gene transcripts. These conidia were more resistant to gliotoxin exposure. Conidial priming could explain the heterogeneity of responses seen within a population of spores exposed to stressors [64]. Each conidium contains a slightly different composition of stored proteins and RNAs based on its location in the conidial chain, which could affect its subsequent response.

In summary, these recent studies have shown that the growth of *A. fumigatus* and *A. nidulans* under stressful environmental conditions can enhance the stress resistance and virulence of the asexual spores (conidia) that they produce. This conidial priming response occurs by transcriptional modulation within the attached conidia that “prepares” them for the subsequent stress [54,55]. It will be of great interest to see if this phenomenon exists in other sporogenic fungi, including other human and plant pathogens. Recently, we showed that when *A. fumigatus* is grown in the presence of subinhibitory concentrations of agricultural triazoles, the conidia it generates are primed and partially protected against subsequent exposure to the medical triazole voriconazole (Harish et al. submitted). Furthermore, these primed conidia develop stable voriconazole resistance at higher rates compared to unprimed conidia. There is already strong evidence that agricultural triazole antifungals generate stable genetic resistance in environmental *A. fumigatus,* through mutations in *cyp51A* that also confer resistance to medical triazoles [65,66]. Therefore, the possibility that agricultural triazoles sprayed on fields at sub-lethal concentrations can prime *A. fumigatus* conidia towards subsequent exposure with medical triazoles or lead to increased resistance to agricultural triazole fungicides in the field is concerning. Further studies performed in an agricultural setting need to be performed to address these concerns.

## 5. Conclusions and Future Directions

The field of fungal priming is in its infancy. It has been demonstrated in only a limited number of species and remains largely descriptive.

The mechanisms of fungal priming remain mostly undiscovered. Numerous questions remain unanswered. For example, does the fungal kingdom contain major shared priming pathways? Do priming mechanisms diverge between the fungal divisions based on their ancestral origins, their lifestyles (saprophytes, plant or human pathogens, mutualists), or their morphology (yeast, hyphal, conidiogenous, etc.)? Do these different lifestyles also affect how long information from a priming stimulus is stored? How can fungi “forget” and “reset” their memories? Can priming affect heritability, for example by increasing the rate of mutations? To what extent do fungi predated by mycophagic animals and microorganisms generate primed spores containing elevated levels of mycotoxins and repellent molecules? Does heat priming generate fungal spores with improved survival when infecting warm-blooded mammals and birds? Answering these questions is not only a fascinating scientific endeavor, but it also has important implications in food security and disease. Fungi are likely undergoing rapid priming and adaptation to the hot, dry, and unpredictable weather patterns caused by global warming, the widespread use of agricultural fungicides, and following anthropogenic introduction to new geographic regions. However, the implications of these processes remain unknown.

## Figures and Tables

**Figure 1 jof-08-00448-f001:**
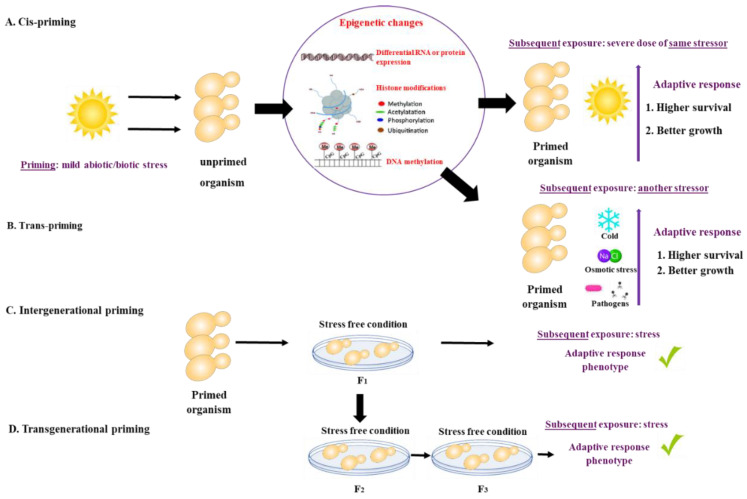
Priming is defined as a time-limited pre-exposure of an organism to a mild stress that leads to an increased adaptive response to subsequent exposures. (**A**) If the initial priming stress and the subsequent exposure are of the same nature, they are referred to as *cis*-priming. (**B**) If priming and exposure differ, they are termed *trans*-priming. Mechanisms of priming include differential RNA or protein expression and storage, histone modifications and DNA methylation. (**C**) Intergenerational priming occurs when the stress memory is observed in the first, stress-free offspring generation, while (**D**) trans-generational priming is observed after more than two stress-free offspring generations.

**Table 1 jof-08-00448-t001:** Animal, plant, and bacterial priming.

Organism	Species	Priming Stress	Exposure Results	Mechanism	References
Mice	*Mus musculus*	Infection with sub-lethal concentrations of the pathogenic mold *A. fumigatus*	80% survival of primed mice, rapid and severe disease onset which cleared after 3 d	Rapid phagocytosis by neutrophiles, elevated levels of the proinflammatory cytokine IL-17	[33]
Mice	*Mus musculus*	Vaccinate mice with liposomal L-mannose protein	Higher survival rate of vaccinated mice after re-infection with *C. albicans*	Elevated production of polyclonal antibodies	[34]
Roundworms	*Caenorhabditis elegans*	Exposure to different stressors: heavy metals, NaCl, fasting.	Increased resistance to fatal oxidative stress which last up to F3	Induction of the transcription factor SKN1 for the oxidative stress response	[13]
Insects	*Galleria mellonela*	Inoculation with sub lethal doses of the pathogen *C.albicans*	Protection against lethal doses of re-infection with same pathogen	Upregulation of antifungal genes (e.g., gallerimycin)	[35]
Insects	*Drosophila Melanogaster*	Inoculation with sublethal doses of the Pathogen *S. pneumoniae*	Protection against lethal doses of re-infection with same pathogen	Higher and efficient phagocytosis	[36]
Plants	*Arabidopsis thaliana*	Exposure to secreted volatiles from a damaged neighboring plant infected with *M*. *separata* larva	Protection against future herbivores	Increased Trypsin inhibitors (TI-plant defense genes) by demethylation of the TI promoter	[23]
Plants	*Arabidopsis thaliana*	Mild osmotic stress (50 mM NaCl)	Higher tolerance to drought stress and extreme osmotic stress (80 mM NaCl)	Histone modification, and increased NaCl transporter (HKT1) induction after second exposure to salt stress	[20]
Plants	*Arrhenatherum elatius*	Dehydration periods of 16 days	Improved photo-protection and higher biomass in second exposure to severe drought	Not known	[21]
Plants	*Triticum aestivum* (winter wheat)	Moderate drought during the vegetative growth period of the plant	Better tolerance to post-anthesis severe drought	Regulating of hormonal levels (e.g., cytokinnines)	[18]
Plants	Soybean seeds	Low to mild concentrations of melatonin	Increased salt tolerance, increased drought tolerance	Upregulation of several genes involved in photosynthesis and sugar metabolism	[24]
Bacteria	*Bacillus subtilis*	Mild heat shock stress (48 °C for 15 min)	Increased tolerance against lethal heat shock stress (53 °C)	Less protein aggregation	[26]
Bacteria	*Escherichia coli*	Subinhibitory concentrations of the antibiotic Ampicillin	Increased resistance to lethal levels of ampicillin, increased resistance to lethal oxidative stress and heat shock stress.	Upregulation of genes involved in higher energy metabolism and more ribosomal production	[31]
Bacteria	*Escherichia coli*	Sublethal doses of AMPs (pexiganan and melittin)	Increased resistance to lethal doses of AMPs (pexiganan and melittin)	Higher amount of colanic-acid capsule in pexiganan-primed cells. Elevated levels of curli fimbriae in melittin-primed cells	[32]
Bacteria	*Listeria monocytogenes*	Exposure to NaCl stress	Increased resistance to the antimicrobial food preservation molecule Nisin	Increased transcript levels of *LiaR*-regulated genes	[29]

**Table 2 jof-08-00448-t002:** Fungal priming.

Species	Priming Stress	Exposure Result	Mechanism	References
*Saccharomyces cerevisiae*	Heat stress (37 °C, 1 h) or osmotic stress (0.7 M, 1 h)	Better tolerance to severe heat shock (47 °C)	Induction of GPDH in osmotic stressed cells, but not in heat-shock stressed cells. Mechanism unknown.	[40]
*Saccharomyces cerevisiae*	Exposure to sub-lethal temperatures	Increased thermotolerance	Mechanism involves the ATPase proton pump	[38]
*Saccharomyces cerevisiae*	Oxidative stress (H_2_O_2_), sub-lethal ethanol stress, cold stress	Barotolerence or high hydrostatic pressure tolerance (HHP)	Upregulation of genes involved in oxidative stress defense response, cell membrane changes	[45]
*Saccharomyces cerevisiae*	Exposure to several mild stressors (acute heat, NaCl, oxidative stress, ethanol)	Resistance to same stressor * (e.g., *p*.heat-*s*.heat), resistance to some subsequent stressors (*p*.NaCl-*s*.NaCl and H_2_O_2_)	Induction of transcription factors Msn2p and/or Msn4p	[43]
*Saccharomyces cerevisiae*	Salt (NaCl) stress	Tolerance to severe oxidative stress (H_2_O_2_)	A role for the nuclear pore component Nup42p	[39]
*Candida albicans*	Mild heat, osmotic or oxidative stress	Heat stressed cells exhibited tolerance to a strong oxidative stress	Slightly increased in HSPs levels (heat shock protein)	[46]
*Candida albicans*	Different concentrations of glucose, low, mild and high	Increased resistance to the antifungal miconazole, increased resistance to osmotic stress and oxidative stress	Upregulation of genes involved in drug resistance, induction of osmotic stress related genes,	[47]
*Metarhizium anisopliae*	Sub-lethal concentrations of heat stress, oxidative stress, osmotic stress and nutritive stress	Increased resistance of *p*.nutritive and *p*.oxidative -*s*.UV-B and *s*.heat. Increased resistance of *p*.heat–*s*.UV-B and heat	Partially established, increased levels of sugars (trehalose and mannitol)	[48]
*Agaricus bisporus*	Application of exogenous riboflavin/vitamin B2	Increased drought resistance	Not established, but changes in transcripts levels was observed	[49]
*Penicillium chrysogenum*	Mild drought stress	Increased resistance to severe drought	Higher β-glucosidase and respiratory activity	[50]
*Rhizopus arrhyzus*	Exposure to sub-lethal concentrations of voriconazole or isavuconazole	In vivo hypervirulence observed by lower survival rate of fruit flies (*D. melanogaster*)	Unknown	[51]
*Rhizopus arrhyzus*	Tornadic shear stress	Hypervirulence in *D. melanogaster* in vivo model	Secreted metabolites and calcineurin-signaling pathway (not fully characterized)	[52]
*Aspergillus fumigatus*	Mild heat stress (37 or 45 °C)	Increased resistance to both oxidative stress and severe heat stress (60 °C)	Increased sugar content (trehalose)	[53]
*Aspergillus fumigatus*	Different environmental stressors: minimal medium, 50 °C, NaCl, +Fe and -Zn	Increased pathogenicity in *D. melanogaster* in vivo model	Not established	[54]
*Aspergillus fumigatus*	Osmotic stress (0.5M NaCl or KCl) Zinc-starved stress, cold (4 °C) or heat (42 °C induced stress	Increased tolerance to oxidative stress, or zinc-stress	Multiple changes in genes transcript levels: *ZapA* (osmotic stress), *Hsp70* and *ZapA* (heat and zinc-starves stress), ergosterol pathway (osmotic stress), gliotoxin secretion (Zinc-starved stress)	[55]

* *p* = priming stress; *s* = subsequent stress.

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
