# Peer review of "Fungal Priming: Prepare or Perish"

_jof, 2022, doi:10.3390/jof8050448_

Round 1

Reviewer 1 Report

Dear authors, since you are writing about spores in fungi, I suggest you look for analogues in plant seeds. There are studies showing that acorns from oaks that were severely defoliated by primary insects (foliar pests) contained more bitter compounds (tannins and phenolic compounds) after germination than the control group. As a result, they were more resistant to possible insect feeding. It is possible that a similar response is seen in pathogenic fungi such as oak powdery mildew Erisiphe alphitoides, which also causes repeated defoliation of trees. This would explain why the greatest damage occurs in stands whose foliage is damaged for the first time compared to stands that are defoliated every year. This also has a practical dimension, because once we have started to protect foliage from insect feeding, we need to continue to apply insecticides, otherwise severe damage or even tree death may occur again. Can priming fungi explain their resistance process to fungicides? Should we abandon chemical pest control in nurseries (agricultural, forestry and ornamental)? Should we accept the associated losses and make a selection of healthy seedlings? If we use pesticides, are not we masking a disease? Plants are infected but asymptomatic, so it is impossible to visually select healthy seedlings. The disease may reappear under favourable conditions after sowing? Therefore, a change in the protection strategy is necessary. The European Commission's Integrated Pest Management Directive addresses this problem.
Finally, how can climate change affect priming and relationships between fungi and plants, such as the roots of fungal crops? When fungi first appeared 450 million years ago, they quickly formed symbiotic relationships with plants, and now it appears that they have even found a way to communicate with each other by transmitting electrical impulses through mycelial hyphae. Will fungi adapt to global warming thanks to priming? So far, they have mainly attacked amphibians and reptiles, rather than warm-blooded organisms such as mammals with a body temperature of around 37°C. If this changes, will many thousands of people die?

Author Response

Reviewer 1

Dear authors, since you are writing about spores in fungi, I suggest you look for analogues in plant seeds. There are studies showing that acorns from oaks that were severely defoliated by primary insects (foliar pests) contained more bitter compounds (tannins and phenolic compounds) after germination than the control group. As a result, they were more resistant to possible insect feeding. It is possible that a similar response is seen in pathogenic fungi such as oak powdery mildew Erisiphe alphitoides, which also causes repeated defoliation of trees. This would explain why the greatest damage occurs in stands whose foliage is damaged for the first time compared to stands that are defoliated every year. This also has a practical dimension, because once we have started to protect foliage from insect feeding, we need to continue to apply insecticides, otherwise severe damage or even tree death may occur again.

REPLY- This is a very interesting idea! We have added this direction in lines 238-239.

Can priming fungi explain their resistance process to fungicides? Should we abandon chemical pest control in nurseries (agricultural, forestry and ornamental)? Should we accept the associated losses and make a selection of healthy seedlings? If we use pesticides, are not we masking a disease? Plants are infected but asymptomatic, so it is impossible to visually select healthy seedlings. The disease may reappear under favourable conditions after sowing? Therefore, a change in the protection strategy is necessary. The European Commission's Integrated Pest Management Directive addresses this problem.

REPLY- It is still not known if priming leads to the development of stable resistance to triazole fungicides in fungi in agricultural settings. We have highlighted this concern on Lines 228-230.

Finally, how can climate change affect priming and relationships between fungi and plants, such as the roots of fungal crops? When fungi first appeared 450 million years ago, they quickly formed symbiotic relationships with plants, and now it appears that they have even found a way to communicate with each other by transmitting electrical impulses through mycelial hyphae. Will fungi adapt to global warming thanks to priming? So far, they have mainly attacked amphibians and reptiles, rather than warm-blooded organisms such as mammals with a body temperature of around 37°C. If this changes, will many thousands of people die?

REPLY- Another excellent idea! We have incorporated it in Lines 242-243

Reviewer 2 Report

The manuscript on "Fungal priming: Prepare or Perish" by Osherov and  Harish is a  review ostensibly on the subject of priming in fungi. In my opinion, Section 1 is well-written and reasonably comprehensive-the first section covering and providing an Introduction to priming.

However Sections 2 , 3 and 4 are "add ons" that are very light and do not cover even one-tenth of the breadth of each of these areas, and therefore should either be removed, as they serve little purpose, or be edited and merged into a general introductory section.

Section 5 on priming in Fungi which is what one would expect from the title, seems to be well reviewed though very descriptive. The molecular mechanisms are not covered in sufficIent depth -is it because there are very few studies on this subject? One such study is by Kandror, O.Bretschneider, N.Kreydin, E.Cavalieri, D. & Goldberg, A. L. (2004). Yeast adapt to near-freezing temperatures by STRE/Msn2,4-dependent induction of trehalose synthesis and certain molecular chaperonesMolecular Cell 13771781.

The limitations in this area are covered in the Discussion Section 6. 

Author Response

Reviewer 2

The manuscript on "Fungal priming: Prepare or Perish" by Osherov and  Harish is a  review ostensibly on the subject of priming in fungi. In my opinion, Section 1 is well-written and reasonably comprehensive-the first section covering and providing an Introduction to priming.

However Sections 2 , 3 and 4 are "add ons" that are very light and do not cover even one-tenth of the breadth of each of these areas, and therefore should either be removed, as they serve little purpose, or be edited and merged into a general introductory section.

REPLY- Sections 2,3,4 have been edited into 8 lines and merged into the general introduction (L60-68).

Section 5 on priming in Fungi which is what one would expect from the title, seems to be well reviewed though very descriptive. The molecular mechanisms are not covered in sufficIent depth -is it because there are very few studies on this subject?

REPLY- Yes, there are few molecular studies, especially in filamentous fungi, the focus of this review.

One such study is by Kandror, O., Bretschneider, N., Kreydin, E., Cavalieri, D. & Goldberg, A. L. (2004). Yeast adapt to near-freezing temperatures by STRE/Msn2,4-dependent induction of trehalose synthesis and certain molecular chaperones. Molecular Cell 13, 771–781.

REPLY- This molecular study is now cited and described (L82-86).

The limitations in this area are covered in the Discussion Section 6.